# Myocarditis in Athletes Recovering from COVID-19: A Systematic Review and Meta-Analysis

**DOI:** 10.3390/ijerph19074279

**Published:** 2022-04-02

**Authors:** Gloria Modica, Massimiliano Bianco, Fabrizio Sollazzo, Emanuela Di Murro, Riccardo Monti, Michela Cammarano, Lorenzo Morra, Francesco Maria Nifosì, Salvatore Francesco Gervasi, Ester Manes Gravina, Paolo Zeppilli, Vincenzo Palmieri

**Affiliations:** 1Sports Medicine Unit, Fondazione Policlinico Universitario A. Gemelli IRCCS, Catholic University, 00168 Rome, Italy; glorymodica@gmail.com (G.M.); massimiliano.bianco@policlinicogemelli.it (M.B.); fabriziosollazzo.md@gmail.com (F.S.); elamanudm@gmail.com (E.D.M.); riccardo.monti1@unicatt.it (R.M.); michela.cammarano01@gmail.com (M.C.); lorenzomorra71993@gmail.com (L.M.); francesconifosi92@gmail.com (F.M.N.); gervasi.salvatore.md@gmail.com (S.F.G.); paolo.zeppilli@unicatt.it (P.Z.); 2Department of Geriatrics, Neurosciences, Orthopaedics and Head and Neck, Università Cattolica del Sacro Cuore-Fondazione Policlinico ‘Agostino Gemelli’ IRCCS, 00168 Rome, Italy; ester.manesgravina@policlinicogemelli.it

**Keywords:** COVID-19, physical activity, myocarditis, athletes, cardiac magnetic resonance

## Abstract

Background: To assess the event rates of myocarditis detected by Cardiac Magnetic Resonance (CMR) in athletes who recovered from COVID-19. Methods: A systematic literature search was performed to identify studies reporting abnormal CMR findings in athletes who recovered from COVID-19. Secondary analyses were performed considering increased serum high sensitivity troponin (hs-Tn) levels and electrocardiographic (ECG) and echocardiographic (ECHO) abnormalities. Results: In total, 7988 athletes from 15 studies were included in the analysis. The pooled event rate of myocarditis was 1% (CI 1–2%), reaching 4% in the sub-group analysis. In addition, heterogeneity was observed (I2 43.8%). The pooled event rates of elevated serum hs-Tn levels, abnormal ECG and ECHO findings were 2% (CI 1–5%), 3% (CI 1–10%) and 2% (CI 1–6%), respectively. ECG, ECHO and serum hs-Tn level abnormalities did not show any correlation with myocarditis. Conclusions: The prevalence of COVID-19-related myocarditis in the athletic population ranges from 1 to 4%. Even if the event rate is quite low, current screening protocols are helpful tools for a safe return to play to properly address CMR studies. Trial registration: the study protocol was registered in the PROSPERO database (registration number: CRD42022300819).

## 1. Introduction

COVID-19, which was first reported in the city of Wuhan, China, has been spreading rapidly worldwide since December 2019 [1,2]. The majority of the SARS-CoV-2 infection symptoms were respiratory in nature (including fever, cough, pharyngodynia, fatigue, and complications related to pneumonia and acute respiratory distress syndrome) [3]. Children and adolescents often experience asymptomatic or paucisymptomatic infection [4]. Cardiac involvement, mainly consisting of myopericarditis, has been described by several authors on patients both with severe disease and with asymptomatic or paucisymptomatic infection [5,6]. Even if the true prevalence of COVID-19-related myocarditis is difficult to establish, due to the lack of the specific diagnostic assessment modalities in early reports, a recent review reported a prevalence between 15% and 27.8% among cases of severe COVID-19 pneumonia [7]. As for cardiac involvement, autopsy reports published so far demonstrated cardiac dilatation, necrosis, lymphocytic infiltration of the myocardium and small coronary vessel microthrombosis as the main cardiac pathological findings [8], with increasing evidence that direct infiltration of SARS-CoV-2 into myocardial cells is possible and frequently accompanied by a strong inflammatory cytotoxic T-cell response [9] and by an interferon-mediated hyperactivation of the innate and adaptive immune system [7]. However, all anatomopathological data may still be influenced by the lack of a uniform post-mortem COVID-19 diagnostic protocol, which has not yet been proposed [10]. The main clinical manifestations of COVID-19 cardiac involvement described so far have been arrhythmias, coronary thrombotic events, acute heart failure and even cardiogenic shock [7].

At the beginning of the pandemic, cases of myocarditis had been frequently described in hospitalized patients, but no information was available regarding cardiac involvement in athletes [5]. As assessed in a recent systematic review, several studies have subsequently investigated the possible cardiovascular complications of COVID-19 infection in young athletes [11]. This because myocarditis is recognized as a cause of sudden death in athletes [12].

For this reason, International Sports Cardiology Expert Panels have developed a set of guidelines on returning to sport in athletes who have had a COVID-19 infection. Cardiopulmonary screening, in this context, has been considered of paramount importance [13,14].

The main purpose of our meta-analysis was to review the current literature and assess the rate of myocarditis events in athletes who were COVID-19 positive, according to the modified Lake and Louise criteria for cardiac magnetic resonance (CMR). Secondly, we also evaluated the event rate of echocardiographic anomalies (in regional kinetics, pericardial effusions), alterations (arrhythmias, ventricular repolarization abnormalities) in the electrocardiogram (ECG) and/or increased serum levels of highly sensitive troponins (hs-TN). Finally, we assessed whether these data were potentially associated with the presence of myocarditis on CMR.

## 2. Materials and Methods

This systematic review and meta-analysis were performed following the Preferred Reporting Items for Systematic reviews and Meta-analysis (PRISMA) Guidelines [15].

### 2.1. Study Search and Selection

Two authors independently screened the title, abstract and full text of the records identified by the search. The following inclusion criteria were applied: (1) cohort studies or observational cross-sectional or case–control studies, and case series that assessed the prevalence of cardiac involvement in athletes who recovered from COVID-19 infection; (2) studies that have used CMR as a cardiovascular imaging technique, possibly combined with ECG and/or echocardiographic (ECHO) studies and/or assessment of serum hs-TN levels; (3) studies that included at least ≥10 athletes who recovered from COVID-19 infection. Publications in languages other than English were not considered and we excluded repeated publication with the same patient cohorts. Any disagreements were resolved through discussion between authors or by involving a third review author.

A systematic search of PubMed, SCOPUS, and Web of Science was performed from 1 January 2020 up to 31 January 2022 for studies which reported cardiac involvement at CMR and, possibly, ECG and/or ECHO abnormalities and/or increased hs-TN levels in athletes who recovered from COVID-19 infection. Following the 16th Bethesda conference, we considered a competitive athlete as an individual who participates in an organized team or in a sport that requires regular competition against others, places a high premium on excellence and achievement, and requires some form of systematic and intense training [16].

The search strategy is summarized in Appendix A.

### 2.2. Data Extraction

The following data were extracted independently by two authors: study characteristics (e.g., study design, country where the study was conducted, number of patients included), patients’ characteristics (e.g., age, sex), symptomatic or asymptomatic athletes, number of athletes who showed signs of cardiac involvement on instrumental examination, and the presence of myocarditis according to the CMR Lake and Louise modified criteria [17].

### 2.3. Data Analysis

For the analysis of myocarditis in athletes, first event rates (ER) for the main outcome were calculated using the mean values, standard deviation (SD), and group size, and pooled using random effects models. Then, a separate meta-analysis was performed, with ER using ECG abnormalities, ECHO abnormalities, and increased hs-TN serum levels. Finally, meta-regression analysis was performed to evaluate the association between abnormal ECG, ECHO or hs-TN serum level, and CMR-detected myocarditis.

The summary ER with 95% CI was estimated for the frequency of myocarditis in athletes. The degree of heterogeneity was assessed using the I^2^ index and Cochrane’s Q. An I^2^ statistic was considered to reflect a low probability (0–25%), moderate probability (26–75%), and high probability (76–100%) of differences beyond chance, as well as a *p*-value, from a Q-test of heterogeneity, less than or equal to 0.05. If the results were homogeneous (I^2^ < 50% and *p* > 0.05), fixed-effects models were used, whereas if were heterogeneous (I^2^ ≥ 50% or *p* ≤ 0.05), random-effects models were used. We also conducted a meta-regression to assess whether in myocarditis event rates were modified by the following patient characteristics: age, sex, country, and COVID-19 symptoms. Evidence of publication bias was examined using Egger’s regression test for funnel skewness, in addition to visual inspection of the funnel plots. Data were summarized using an inverse variance-weighted meta-analysis and a 2-sided *p*-value less than or equal to 0.05 was considered significant. Statistical analysis was performed using Prometa, version 3.

### 2.4. Assessment of Risk of Bias

The bias risk assessment was carried out using the forms included in the “Joanna Briggs Institute (JBI) Critical Appraisal tools checklists”. Appendix A show the results (see Appendix A).

## 3. Results

### 3.1. Study Selection

The first screening was carried out on 293 articles identified by title and abstract through the systematic database search (Figure 1 and Appendix A). After removing 211 duplicates, 82 full-text records were evaluated, after which a further 55 articles were excluded (21 were reviews; 19 were expert consensus, recommendations, and/or guidelines; 8 were letters; 6 were case reports; and 1 errata corrige). Of the remaining 27 journal articles evaluated as full text, 1 was excluded because the language was not English; 1 because of analyzing the same population; 3 because the populations studied were not athletes; 1 because the item was not myocardial injury; and 6 because CMR Lake and Louise criteria were missing, or it was not specified whether they had been used (CMR was not performed). Finally, 15 [18,19,20,21,22,23,24,25,26,27,28,29,30,31,32] studies were included for a total of 7988 athletes (29.9% female), but only 2390 athletes had a CMR study. Agreement among reviewers was excellent, with a kappa statistic of 0.90.

### 3.2. Study Characteristic

The size of the included studies ranged from 12 to 3018 athletes (Table 1). Study design was 1 case series [27], 1 case–control study [24], 10 cohort studies [19,21,22,23,25,26,28,30,32,33], and 3 cross-sectional observational studies [18,29,31]. Nine studies were conducted in North America and six in Europe.

The overall mean age was 20.0 ± 2.88 years. CMR was performed after a mean of 22.0 ± 7.14 days from the first positive PCR test for SARS-CoV2. Two studies [21,25] did not report the athletes’ age and five [24,25,28,30,32] studies did not report the time from the diagnosis when CMR was done. ECG and ECHO were not reported only in one study [26] and bloodwork for hs-TN was not performed only in one study [30]; however, we decided to include them in our meta-analysis, as these instrumental findings were secondary outcomes.

Overall, the ER of myocarditis in 7988 athletes was 1% (95% CI, 1–2%; *p*-value < 0.001; Figure 2A–C), with an I^2^ of 43.8. The funnel plot in Appendix A shows the publication bias. Egger’s linear regress test was not significant (*p*-value = 0.98).

After that, we made a sub-group analysis considering only the 2390 athletes who were studied with CMR (Figure 2B). Data showed an ER of myocarditis of 4% (CI 1–6%; *p* < 0.001), with an I^2^ of 39.4. The publication bias by Egger’s linear regress test was significant (*p*-value 0.05).

For this reason, we tested the hypothesis that studies [19,20,26,27,31] in which CMR was performed in the whole COVID-19-positive athlete population had more homogeneous results. Figure 2C shows data of this analysis: in this subgroup of 389 athletes, the ER of myocarditis was 2% (CI 1–4%; *p* < 0.001), with an I^2^ of 39.4. The publication bias by Egger’s linear regress test was not significant (*p*-value = 0.61).

Furthermore, we analyzed the presence of CMR abnormalities even without meeting the Lake and Louise modified criteria for the diagnosis of myocarditis. Data showed an ER of CMR abnormalities of 4% (CI 1–9%; *p* < 0.001), with an I^2^ of 96.3. The publication bias by Egger’s linear regress test was not significant (*p*-value = 0.93).

When we considered the instrumental or laboratory signs of myocardial involvement, the data showed an ER for ECG abnormalities of 3% (CI 1–10%; *p* value < 0.001), for ECHO abnormalities of 2% (CI 1–6%; *p*-value < 0.001), and for increased level of serum hs-TN of 2% (CI 1–5%; *p* = 0.000), without a significant publication bias for all (*p* = 0.28, *p* = 0.54 and *p* = 0.58, respectively).

A meta-regression analysis using publication year, study design, age, sex, and presence of symptoms as moderators did not show any significant findings (*p* > 0.05).

Finally, we evaluated the possible correlation between myocarditis detected at CMR and ECG, ECHO, and hs-TN elevation, which did not demonstrate any significant correlation.

## 4. Discussion

### 4.1. COVID-19 Myocarditis in General People and in Athletes

At the beginning, cases of COVID-19-related myocarditis were described only in hospitalized patients [5]. A recent meta-analysis investigated the prevalence of myocarditis in both patients and a subgroup of 316 athletes, post COVID-19 infection. In the subgroup analysis, the aggregate prevalence of CMR abnormalities and myocarditis was significantly higher in non-athletes than in athletes (62.5% versus 17.1% and 23.9% versus 2.5%, respectively) [6]. One explanation for this observed difference could be the presence of comorbidity in the group of patients compared to athletes. To support this, a systematic review by Ahmed M. A. Shafi et al. confirmed that hypertension and diabetes are important determinants of cardiovascular events during COVID-19 infection [34].

Our meta-analysis, conducted only in athletes (*n* = 7988), showed an ER of myocarditis of 1% (95% CI, 1–2%; *p* < 0.001), an ER of myocarditis of 4% (CI 1–6%; *p* < 0.001) if we made a sub-group analysis in all athletes who performed CMR studies (*n* = 2390), and an ER of myocarditis of 2% (CI 1–4%; *p* < 0.001) in studies in which CMR was performed directly in the whole COVID-19-positive population (*n* = 389) [19,20,26,27,31]. These results are probably due to the fact that there is a wide heterogeneity in studies design, namely a few studies made CMR in all positive athletes [19,20,21,26,27,31], whereas other researches made a CMR only on a clinical suspicion basis [18,21,22,23,24,25,28,29,30,31,32]. For example, in the study by Curt Daniels et al. [21], four distinct approaches were used to refer athletes with recent COVID-19 infection to CMR: the first approach was based on the presence of symptoms; the second, on the combined use of ECG parameters, ECHO, hs-TN test, and other abnormalities; the third approach was based on abnormalities in at least one of ECG, ECHO, hs-TN test, or other anomalies; and the last was the direct use of CMR. On the other side, Cavigli et al. [30] used a strict diagnostic algorithm, so that CMR had been performed only in 3 out of 571 screened athletes (0.5%). This could also explain the relatively wide range of I^2^ (from 39.4 to 43.8) we detected among all our analysis.

### 4.2. Pathophysiology of the COVID-19 Myocarditis

The pathophysiology of myocardial involvement during COVID-19 infection is still debated. Some hypotheses include direct damage to cardiomyocytes, systemic inflammation, myocardial interstitial fibrosis, interferon-mediated immune response, exaggerated cytokine response by T helper cells type 1 and 2, coronary plaque destabilization, hypoxia [35], and molecular mimicry [36], among others. However, the two main theories at present are (1) the direct role of angiotensin-converting enzyme 2 (ACE2) receptors; and (2) a hyperimmune response [37]. Teresa Castiello et al. [38], in their recent systematic review that included 38 case reports, showed only in one case the presence of SARS-CoV-2 in endomyocardial biopsy (EBM). In the other cases, histology showed inflammation of the myocardium with a predominance of macrophages, whereas myocyte necrosis was limited.

### 4.3. Why Is the Diagnosis of Myocarditis in Athletes Important?

Myocarditis plays an important role in the pathogenesis of sudden cardiac death (SCD) in athletes [39]. Physical exertion is probably a trigger for dangerous arrhythmias and may have the potential to further propagate myocardial damage in athletes with myocarditis [40]. In adjunct, exercise has an important impact on immune function and may therefore lead to changes in the biological response of athletes to myocarditis. Exercise of moderate intensity can significantly improve the immune response [41], whereas intense exercise (and a lack in restoration) can lead to a dramatic decline in immune function [42]. Myocarditis represents a potential cause of SCD in athletes: in acute myocarditis, myocardial inflammation represents an arrhythmogenic substrate that predisposes patients to ventricular arrhythmias (VAs); in chronic myocarditis, on the other hand, myocardial fibrosis promotes VAs through the creation of re-entry circuits around the myocardial scar [43]. For these reasons, current guidelines recommend a rest period of three to six months after the diagnosis of myocarditis [44].

Nowadays, as stated above, it is well known that COVID-19 may predispose to myocarditis, even in athletes, and this is the reason that leads International Sports Cardiology authors and societies to propose screening protocols aimed to assess the cardiopulmonary system in athletes who recovered from COVID-19 [14,45,46,47].

### 4.4. Cardiac Magnetic Resonance’ Abnormalities in Athletes

If we consider all cases in which CMR abnormalities did not fit the standards for myocarditis diagnosis according to the Lake and Louise modified criteria, data showed an ER of CMR abnormalities of 4% (CI 1–9%; *p* < 0.0001). These findings must be considered with caution [48] because it is well known that some CMR findings in athletes may be misleading and their clinical relevance it is not clarified to date. For example, a recent meta-analysis [49] showed a higher prevalence of LGE among endurance athletes than gender-matched controls (odds ratio of 7.20). Moreover, the presence of late-gadolinium enhancement (LGE) on CMR is gaining a potential role as a strong independent predictor of SCD, all-cause mortality, and cardiac mortality [43]. To date, observational data suggest that LGE in the midwall and septal segments carries the highest risk of SCD in the context of both reduced and preserved left ventricular ejection fraction (LVEF), independently from clinical symptoms, and LGE seems to be superior to other prognostic factors such as LVEF, left ventricular end diastolic volume, or NYHA functional class [43]. Zorzi et al. demonstrated that isolated, non-ischemic left ventricular LGE may be associated with life-threatening arrhythmias and SCD in athletes and that, due to its subepicardial/myocardial location, left ventricular scarring is often not detected by echocardiography [50]. Further interpretation of these findings, even in SARS-CoV-2 infection, needs follow-up data, which are not always provided in all available studies: in fact, only a few studies [19,22,27,28,31] reported short-term follow-up data with a control CMR, with a small number of athletes involved. As for arrhythmias, the issue is still present: only three studies [24,30,32] evaluated the presence of arrhythmias after COVID-19 (two of them comparing it with a pre-COVID ECG) and none had an arrhythmic follow-up through time.

### 4.5. Instrumental Abnormalities in Athletes Who Recovered from COVID-19

The secondary outcomes of our study (ECG abnormalities, ECHO abnormalities, increased serum levels of hs-TN) showed a non-negligible event rate (ER of 3%, CI 1–10%; ER of 2%, CI 1–6%; ER of 2%, CI 1–5%, respectively). However, studies conducted to date have very little follow-up data, and only two studies [24,28] assessed arrhythmic risk. Finally, almost all studies assessed hs-TN levels, but none showed a correlation with the presence of cardiovascular abnormalities. This finding confirmed what was pointed out by a recent systematic review [11], thus confirming that hs-TN changes may be related to a broad set of clinical reason. A recent review [51] showed that normal ranges in athletes are still poorly defined, and further studies are needed to assess the normal TN value in athletes.

### 4.6. SARS-CoV-2 Vaccination and Myocarditis or Myopericarditis

In a recent large cohort study by A. Husby et al. [52] the association between mRNA-1273 vaccination and an increased rate of myocarditis or myopericarditis compared to unvaccinated individuals was confirmed; an increased rate of myocarditis or myopericarditis was also observed among female individuals with BNT162b2 vaccination. However, this study, in contrast to the Israeli [53,54,55] study and the USAstudy [56], showed an overall low absolute rate of myocarditis or myopericarditis cases after SARS-CoV-2 mRNA vaccination among female participants and younger age groups. It also highlighted that clinical outcome after myocarditis or myopericarditis events are predominantly mild, providing evidence to support the overall safety of SARS-CoV-2 mRNA vaccines.

### 4.7. Limitations

The data collected in this meta-analysis considered the SARS-CoV-2 variants known at the time of publication of the included articles; therefore, we do not know what cardiovascular effects the other variants, and specifically omicron variant, could have. We are also aware that the athletes’ vaccinal data were not collected, and that there is a not-negligible correlation between COVID-19 vaccines and myocarditis [57].

## 5. Conclusions

The prevalence of COVID-19-related myocarditis in the athletic population varies in different studies, but the overall estimate rate is quite low, around 1%. However, if CMR study is requested based on a clinical-instrumental suspicion, the estimate rate is around 4%. Even if the rate remains relatively low, as myocarditis remains one of the causes of SCD in athletes, recognizing and treating it is of paramount importance [14,45,46]. Current screening protocols are helpful tools for a safe return to play in athletes who recovered from COVID-19 and, guided by a good clinical practice, should lead a decision in CMR studies. As for the relationship between CMR and other laboratory or instrumental abnormalities (ECG, ECHO, and hs-TN), there is not any strong association between these parameters.

## Figures and Tables

**Figure 1 ijerph-19-04279-f001:**
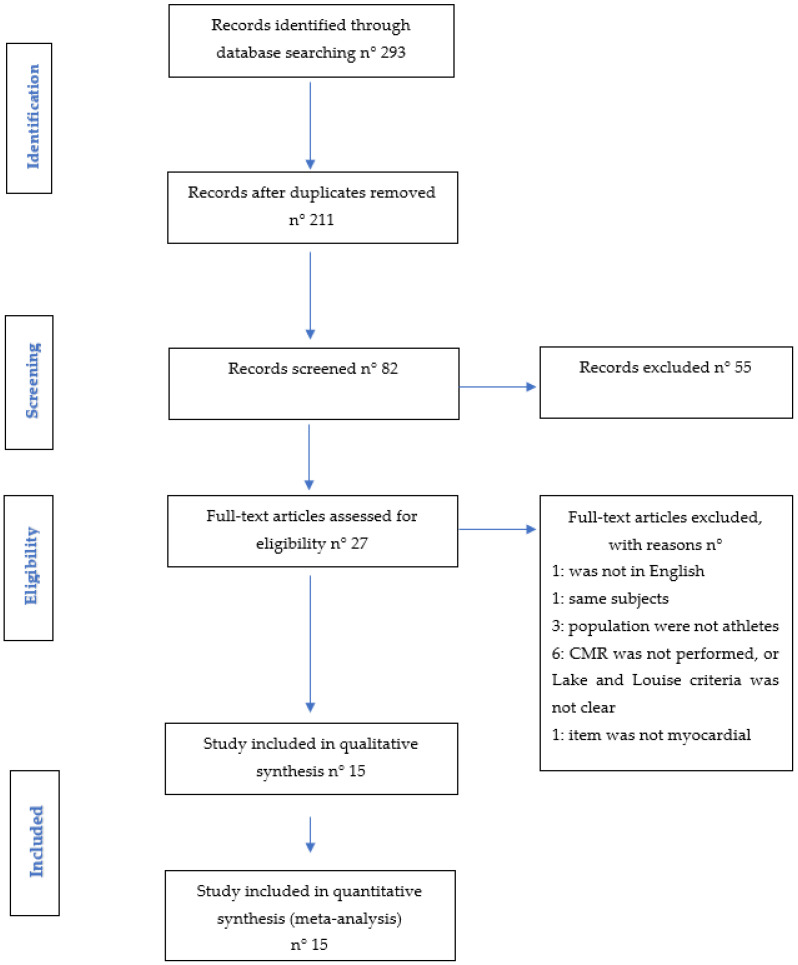
PRISMA’s flow diagram.

**Figure 2 ijerph-19-04279-f002:**
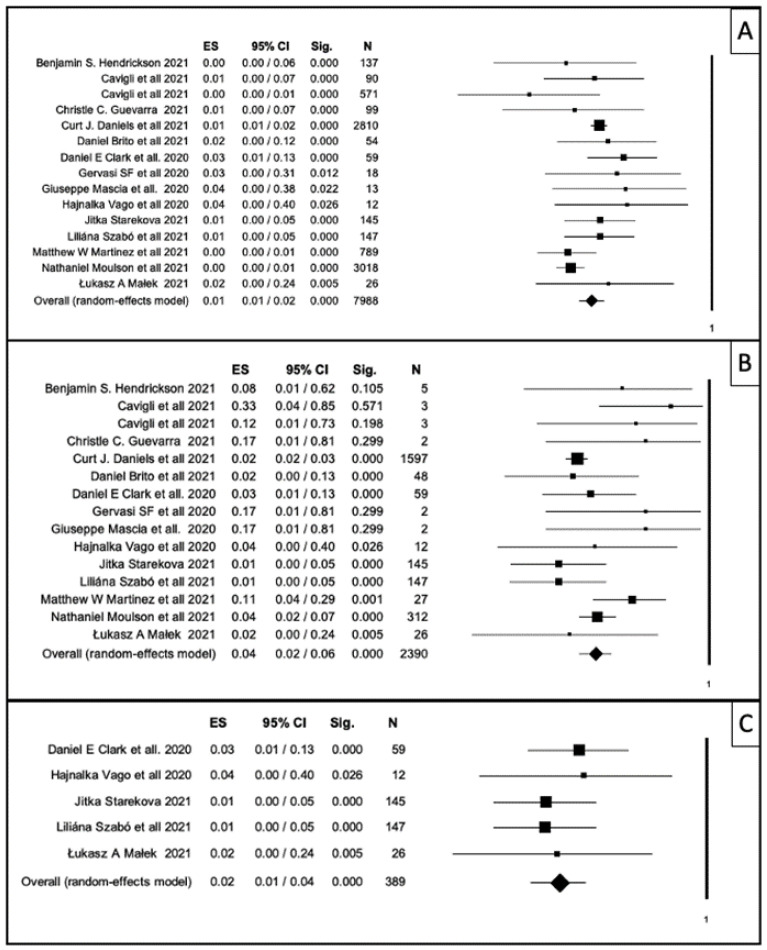
Forest plot: horizontal axis represents 95% confidence interval; rhombuses and squares represent the weight of the study in relation to the total. (**A**) Forest plot showing myocarditis event rate in the total sample of athletes (*n* = 7988) who recovered from COVID-19 infection. (**B**) Forest plot showing myocarditis event rate only in the subgroup of athletes (*n* = 2390) who were studied with CMR. (**C**) Forest plot showing myocarditis rate in studies in which all athletes who recovered from COVID-19 were studied with CMR. This subgroup included 389 athletes.

**Table 1 ijerph-19-04279-t001:** Study characteristics.

Authors	Study Design	Center	Continent	Athletes Tested COVID-19 Positive (*n*)	Mean Age	Female (*n*)	CMR—Mean Days after Infection
Gervasi SF et al. 2020	Case–control	Single Center	Europe	18	23.2	0	n.a.
Daniel E Clark et al. 2020	Cohort study	Single Center	North America	59	20	37	21.5
Hajnalka Vago et al. 2020	Cohort study	Single Center	North America	12	23	10	17
Giuseppe Mascia et al. 2020	Cohort study	Single Center	Europe	13	n.a.	0	n.a.
Curt J. Daniels et al. 2021	Cohort study	Multicenter	North America	2810	n.a.	931	22.5
Daniel Brito et al. 2021	Cross-sectional observational study	Single Center	North America	60	19	8	27
Benjamin S. Hendrickson et al. 2021	Cohort study	Single Center	North America	137	20	44	16.0
Jitka Starekova et al. 2021	Case series	Single Center	North America	145	20	37	15
Matthew W Martinez et al. 2021	Cross-sectional observational study	Multicenter	North America	789	25	12	19
Nathaniel Moulson et al. 2021	Cohort study	Multicenter	North America	3018	20	966	37.7
Łukasz A Małek et al. 2021	Cohort study	Single Center	Europe	26	24	21	32
Cavigli et al. 2021	Cohort study	Multicenter	Europe	90	24	26	n.a.
Cavigli et al. 2021	Cohort study	Multicenter	Europe	571	14.3	221	n.a.
Liliána Szabó et al. 2021	Cross-sectional observational study	Single Center	Europe	147	23.0	53	32
Christle C. Guevarra 2022	Cohort study	Single Center	North America	99	19.9	31	n.a.

Legend: n.a. = not available.

## Data Availability

The data presented in this study are openly available on the studies referenced in the figures, and individual data of each can be consulted in the original manuscripts.

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
