# Peer review of "Myocarditis in Athletes Recovering from COVID-19: A Systematic Review and Meta-Analysis"

_ijerph, 2022, doi:10.3390/ijerph19074279_

Round 1
Reviewer 1 Report
The intent of this study is noteworthy and of importance to the athletic population. However, this is not made clear in the Introduction. The incidence (and prevalence) of myocarditis must be recognized in the non-athletic population to gain a better understanding why this study was initiated, and whether the athletic population is affected (or protected) from COVID. If the incidence of cardiac involvement is the same in athletes versus non-athletes (See paper by A. Husby in BMJ, 2021), then what do we gain from this study?
In the Introduction, lines 32 forward, the authors can expand (for readership) on the effects of COVID-19 on heart function (dimensions, EF, etc.) and electrical activity (type of disturbance, etc).
In the Discussion, a brief mention of the protective effects of vaccines (Pfizer and Moderna) on cardiac involvement is warranted.
Author Response
Thank you for your comment that will be very helpful in increasing the quality of our manuscript.
We have added some information on myocardial involvement in Covid-19 patients in the introduction. In addition, we have implemented the introductory part of our study with prevalence data of Covid-19 myocarditis in general population, consisting, mainly of hospitalized patients so to have a better understanding throughout the paper of the differences with athletic subjects.
One of the objectives of our study is to provide, by means of a cumulative statistical analysis of the studies published to date, the rate of myocarditis events in the athletic population.
We also introduced in our discussion, as you suggested, another paragraph on SARS-CoV-2 vaccination and myopericarditis.
Reviewer 2 Report
Dear authors,
it's a pleasure for me to collaborate on this peer-review and congrats your interesting work.
Here some considerations:
[line 39]: please better explain the pathological involvement of the heart by citing articles such as:
Maiese A, Manetti AC, La Russa R, Di Paolo M, Turillazzi E, Frati P, Fineschi V. Autopsy findings in COVID-19-related deaths: a literature review. Forensic Sci Med Pathol. 2021 Jun;17(2):279-296. doi: 10.1007/s12024-020-00310-8. Epub 2020 Oct 7. PMID: 33026628; PMCID: PMC7538370.
Gauchotte G, Venard V, Segondy M, Cadoz C, Esposito-Fava A, Barraud D, Louis G. SARS-Cov-2 fulminant myocarditis: an autopsy and histopathological case study. Int J Legal Med. 2021 Mar;135(2):577-581. doi: 10.1007/s00414-020-02500-z. Epub 2021 Jan 3. PMID: 33392658; PMCID: PMC7779100.
Maiese A, Frati P, Santoro P, Manetti AC, La Russa R, Di Paolo M, Turillazzi E, Fineschi V. Myocardial Pathology in COVID-19-Associated Cardiac Injury: A Systematic Review. Diagnostics (Basel). 2021 Sep 8;11(9):1647. doi: 10.3390/diagnostics11091647.
[Line 50]: in your discussion, you evaluate the importance of myocarditis in athletes, but you don't include this topic among your aims, please consider to add it on your aims [it's not necessary].
[Figure 2]: Could you describe what is on the horizontal axis of the forest plot?
[line 225]: please, explain better from your analysis, if covid19 myocarditis is a risk factor [dependent or independent] for cardiac sudden death.
Congratulation for you brilliant analysis, good work
Author Response
Thank you for your comment.
We have added (line 39) histopathological information on the cardiac involvement of COVID-19, as you requested, in the introductory part of our paper citing the interesting articles you suggested.
As you can read from the last paragraph of our introduction, we have clearly defined the main and secondary outcomes of our study.
We have also explained, in Fig. 2, the meaning of the horizontal axis and the rhombuses and squares of the Forest Plot.
Regarding the comment to line 225, to date we have no long-term follow-up data from the studies conducted. However, the short-term follow-up data, provided by the studies included in our analysis, do not show cases of sudden deaths from COVID-19-related myocarditis among athletes. This does not, however, eliminate the possibility, as known in the literature, that sudden death from sport is a possible event during myocarditis.
Round 2
Reviewer 1 Report
No comments. The authors have done a good job addressing my comments/suggestions.